# Hierarchical classification at multiple operating points

**Jack Valmadre**
Australian Institute for Machine Learning, University of Adelaide
`jack.valmadre@adelaide.edu.au`

## Abstract

Many classification problems consider classes that form a hierarchy. Classifiers that are aware of this hierarchy may be able to make confident predictions at a coarse level despite being uncertain at the fine-grained level. While it is generally possible to vary the granularity of predictions using a threshold at inference time, most contemporary work considers only leaf-node prediction, and almost no prior work has compared methods at multiple operating points. We present an efficient algorithm to produce operating characteristic curves for any method that assigns a score to every class in the hierarchy. Applying this technique to evaluate existing methods reveals that top-down classifiers are dominated by a naïve flat softmax classifier across the entire operating range. We further propose two novel loss functions and show that a soft variant of the structured hinge loss is able to significantly outperform the flat baseline. Finally, we investigate the poor accuracy of top-down classifiers and demonstrate that they perform relatively well on unseen classes. Code is available online at https://github.com/jvlmdr/hiercls.

## 1 Introduction

Many classification problems involve classes that can be recursively grouped into a hierarchy of progressively larger superclasses. This hierarchy can be represented by a directed graph where the nodes are classes and the edges define a superset-of relation. Knowledge of the hierarchy can be useful in many different respects. For example, mistake severity can be quantified using distance in the graph [9, 2], classifiers can make predictions at a coarse level to avoid an error at the fine-grained level [11, 39], classes with few labels in a long-tailed distribution can benefit from the examples of similar classes [39], and a cascade can be used to reduce inference time [26, 16, 14].

Whereas many recent works consider only leaf-node prediction, we are interested in the setting where the classifier may predict any class in the hierarchy, including internal nodes. In this setting, prediction involves an inherent trade-off between specificity and correctness: more general predictions contain less information but have a greater chance of being correct. The trade-off can typically be controlled using an inference threshold, analogous to the trade-off between sensitivity and specificity in binary classification or that between precision and recall in detection. However, while it is standard to evaluate these problems using trade-off curves, most works on hierarchical classification consider only a single operating point. It is important to consider the full trade-off curve in order to ensure a fair comparison and enable the selection of classifiers according to design specifications. This paper presents an efficient algorithm to obtain trade-off curves for existing hierarchical metrics.

The proposed technique is subsequently used to compare the trade-offs realised by different methods. Given the effectiveness of deep learning, we focus on loss functions for training differentiable models. Experiments on the iNat21 [36] and ImageNet-1k [10] datasets for image classification reveal that a naïve flat softmax classifier *dominates* the more elegant top-down classifiers, obtaining better

36th Conference on Neural Information Processing Systems (NeurIPS 2022).

accuracy at any operating point. We further introduce a soft structured-prediction loss that dominates the flat softmax.

While it was already known that top-down approaches provide worse *leaf-node* predictions than a flat softmax [31, 2], we did not expect this to hold for non-leaf predictions. We hypothesise that the top-down approaches obtain worse accuracy than the "bottom-up" flat softmax because the coarse classes can be highly diverse, and thus better learned by a union of distinct classifiers when the fine-grained labels are available. To support this hypothesis, we show that training a flat softmax classifier at a lower level and testing at a higher level provides better accuracy than training at the higher level. Finally, we consider a synthetic *out-of-distribution* problem where a hierarchical classifier that explicitly assigns scores to higher-level classes may be expected to have an advantage.

The key contributions of this paper are as follows.

- We introduce a novel, efficient technique for evaluating hierarchical classifiers that captures the full trade-off between specificity and correctness using a threshold-based inference rule.
- We propose two novel loss functions, soft-max-descendant and soft-max-margin, and entertain a simplification of Deep RTC [39] which we refer to as parameter sharing (PS) softmax. While soft-max-descendant is ineffective, soft-max-margin and PS softmax achieve the best results.
- We conduct an empirical comparison of loss functions and inference rules using the iNat21-Mini dataset and its hierarchy of 10,000 species. The naïve softmax is found to be surprisingly effective, and the simple threshold-based inference rule performs well compared to alternative options.
- We evaluate the robustness of different methods to unseen leaf-node classes. The top-down methods are more competitive for unseen classes, while PS softmax is the most effective.

## 2 Related work

There are several different types of hierarchical classification problem. In the terminology of Silla and Freitas [33], we consider problems with tree structure, Single-Path Labels (SPL) and Non-Mandatory Leaf-Node Prediction (NMLNP). *Tree structure* means that there is a unique root node and every other node has exactly one parent, *Single-Path Labels* means that a sample cannot belong to two separate classes (unless one is a superclass of the other) and *Non-Mandatory Leaf-Node Prediction* means that the classifier may predict any class in the hierarchy, not just leaf nodes. This work assumes the hierarchy is known and does not consider the distinct problem of learning the hierarchy.

**MLNP with deep learning.** Several recent works have sought to leverage a class hierarchy to improve leaf-node prediction. Wu et al. [38] trained a softmax classifier by optimising a "win" metric comprising a weighted combination of likelihoods on the path from the root to the label. Bertinetto et al. [2] proposed a similar Hierarchical Cross-Entropy (HXE) loss comprising weighted losses for the conditional likelihood of each node given its parent, as well as a Soft Label loss that generalised label smoothing [27] to incorporate the hierarchy. Karthik et al. [20] demonstrated that a flat softmax classifier is still effective for MLNP and proposed an alternative inference method, Conditional Risk Minimisation (CRM). Guo et al. [18] performed inference using an RNN starting from the root node. Other works have proposed architectures that reflect the hierarchy (*e.g.* [41, 1, 43]). This paper seeks to compare loss functions under a simple inference procedure.

**Non-MLNP methods.** Comparatively few works have entertained hierarchical classifiers that can predict arbitrary nodes. Deng et al. [11] highlighted the trade-off between specificity and accuracy, and sought to obtain the most-specific classifier for a given error rate. Davis et al. [7, 8] re-calibrated a flat softmax classifier within a hierarchy using a held-out set and performed inference by starting at the most-likely leaf-node and climbing the tree until a threshold was met, comparing methods at several fixed thresholds. Wu et al. [39] proposed the Deep Realistic Taxonomic Classifier (Deep RTC), which obtains a score for each node by summing over those of its ancestors [32] and whose loss is evaluated at random cuts of the tree. They perform inference by traversing down from the root until a score threshold is crossed (by default, zero). In the YOLO-9000 detector, Redmon and Farhadi [31] introduced a conditional softmax classifier that resembled the efficiency-driven approach of Morin and Bengio [26]. The model outputs a concatenation of logit vectors that each parametrise (via softmax) the conditional distribution of children given a parent. It was noted to provide elegant degradation but the hierarchical predictions were not rigorously evaluated. Brust and Denzler [4] generalised the conditional approach to multi-path labels in a DAG hierarchy by replacing the softmax with a sigmoid for each class given *each* of its parents. Inference was performed by seeking the class

with the maximum likelihood excluding its children, which may predict non-leaf nodes. Most of the above loss functions will be compared in the main evaluation.

**Hierarchical metrics.** There are many ways to measure the accuracy of hierarchical classifiers [6, 21]. For the specific case of NMLNP, it is important to consider metrics for both correctness and specificity. The Wu-Palmer metrics [40] measure precision and recall using the depth of the Lowest Common Ancestor (LCA). Deng et al. [11] proposed to measure specificity using information content to account for an imbalanced tree, and Zhao et al. [42] modified the Wu-Palmer metrics to use information rather than depth. These are the metrics that we adopt, although our technique can be applied to construct curves for any simply metric. Examples of other metrics include the fraction of examples receiving non-root classifications [7] and the fraction of true-negative leaf nodes for correct predictions [39].

**Structured prediction.** The max-margin structured-prediction loss has seen historical use for hierarchical classification with linear SVMs, originally for document classification [5] and later to achieve efficient inference [34]. More recently, a soft max-margin loss [29, 15] has been used to train deep models for sequence-to-sequence learning [13] and long-tail classification [25]. We believe this paper is the first to recognise its utility for hierarchical classification with deep learning.

# 3  Problem definition

## 3.1  Class hierarchy

Let $\mathcal{Y}$ be the set of classes, including both leaf nodes and their superclasses. The hierarchy is defined by a tree with edges $\mathcal{E} \subset \mathcal{Y}^2$. The edges define a transitive, non-strict superset relation $\supseteq$ over the classes. It is helpful to think of the classes as sets of samples. A sample $x$ that belongs to class $y$ also belongs to the ancestors (superclasses) of $y$; that is, $x \in y$ and $y \subseteq y'$ implies $x \in y'$. Hierarchical classification can be seen as multi-label classification, since samples belong to multiple classes simultaneously. However, samples cannot belong to *arbitrary* subsets of $\mathcal{Y}$, as belonging to one class implies belonging to its superclasses. Furthermore, we consider only class hierarchies in which siblings are *mutually exclusive*; that is, a sample $x$ can only belong to two classes $y$ and $y'$ if one is a superclass of the other. The problem is therefore to assign a single label $y \in \mathcal{Y}$ to an example, with this label signifying membership in class $y$ and all of its superclasses. While any superclass of the ground-truth label is deemed to be a correct classification, it is preferable for the classifier to predict the most-specific correct label.

We briefly introduce our notation for the tree. Let $\mathbf{r} \in \mathcal{Y}$ be the unique root node, let $\pi(y) \in \mathcal{Y}$ be the unique parent for every node $y \in \mathcal{Y} \setminus \{\mathbf{r}\}$, let $\mathcal{C}(y) \subseteq \mathcal{Y}$ be the children of node $y$, let $\mathcal{L} \subseteq \mathcal{Y}$ be the set of leaf nodes, let $\mathcal{A}(y) \subseteq \mathcal{Y}$ be the ancestors of node $y$, and let $\mathcal{D}(y) \subseteq \mathcal{Y}$ be the descendants of node $y$. We define the ancestors and descendants inclusively, such that $\mathcal{D}(y) \cap \mathcal{A}(y) = \{y\}$. It will also be useful to introduce $\mathcal{L}(y) = \mathcal{L} \cap \mathcal{D}(y)$ to denote the set of leaf descendants of node $y$ and $\mathcal{S}(y) = \mathcal{C}(\pi(y))$ to denote its siblings. The superset relation over classes is equivalent to the ancestor relation in the tree; that is, $u \supseteq v$ iff $u \in \mathcal{A}(v)$.

We focus on models that map a sample $x$ to a conditional likelihood $p(y|x)$ over all labels in the hierarchy. If we consider $\mathcal{Y}$ with its superset relation and mutually exclusivity of siblings as an event space, then a function $p : \mathcal{Y} \to [0, 1]$ must satisfy

$$\forall u \in \mathcal{Y} : \quad p(u) \geq \sum_{v \in \mathcal{C}(u)} p(v) \tag{1}$$

to be a probability function on $\mathcal{Y}$. If this holds with equality, we say that the children are exhaustive. We generally expect that the root node is uninformative $p(\mathbf{r}) = 1$.

## 3.2  Metrics

Metrics $M(y, \hat{y})$ measure the quality of predicted label $\hat{y}$ for ground-truth label $y$. When making non-leaf predictions, it is important to capture both correctness and specificity, either using a pair of metrics or a single combined metric. The simplest such pair are the binary metrics

$$\text{Correct}(y, \hat{y}) = [\hat{y} \supseteq y] , \qquad\qquad \text{Exact}(y, \hat{y}) = [\hat{y} = y] . \tag{2}$$

If the ground-truth label is not a leaf node, then it is possible for the predicted label to be below it, $\hat{y} \subset y$. This is not considered an error, and we replace $\hat{y} \leftarrow y$ where this occurs.

Since the EXACT metric is often too strict, it is useful to introduce a non-binary measure of label specificity. Two popular choices are the depth of the node $d(y)$ and its information content $I(y) = -\log p(y) = \log|\mathcal{L}| - \log|\mathcal{L}(y)|$ (assuming a uniform distribution over the leaf nodes [11, 42]). We prefer information as it better accounts for an imbalanced tree; in a perfect $k$-ary tree, $I(y) \propto d(y)$. Rather than directly measure the specificity of the predictions [11], the specificity measure can be used to define precision and recall metrics [42] using the Lowest Common Ancestor (LCA):

$$\text{Recall}(y, \hat{y}) = \frac{I(\text{lca}(y, \hat{y}))}{I(y)} \,, \qquad \text{Precision}(y, \hat{y}) = \frac{I(\text{lca}(y, \hat{y}))}{I(\hat{y})} \,. \qquad (3)$$

Correct and Precision are measures of correctness (decreasing with depth), whereas Exact and Recall are measures of specificity (increasing with depth). Each type should not be used without the other.

## 4 Inference and loss functions

We adopt the paradigm that a differentiable model $f$ with parameters $\phi$ maps an input $x$ to a real vector $\theta = f(x, \phi)$ that parametrises the conditional likelihood $p(\cdot|x) = q(\theta) \in [0, 1]^{\mathcal{Y}}$. Learning will be performed using Stochastic Gradient Descent (SGD) to minimise the expected value of a loss function $\ell(y, \theta)$ on a training set. Inference will be performed using a function $\xi : [0, 1]^{\mathcal{Y}} \to \mathcal{Y}$ to obtain $\hat{y} = \xi(q(\theta))$. We consider the training method to be defined by the pair $(\ell, q)$ and the inference method to be defined by $\xi$. The dimension of $\theta$ depends on the training method.

### 4.1 Inference functions

In standard flat classification, inference simply selects the most likely class $\xi(p) = \arg\max_{y \in \mathcal{Y}} p(y)$. However, in the hierarchical setting, this would always select the uninformative root node. We can consider **leaf** inference $\xi(p) = \arg\max_{y \in \mathcal{L}} p(y)$, however this will never select an internal node. We propose **confidence threshold** inference, taking the maximum-information node

$$\xi_\tau(p) = \arg\max_{y \in \mathcal{Y}} I(y) \quad \text{subject to} \quad p(y) > \tau \,. \qquad (4)$$

When $\tau \geq 0.5$ and $p$ satisfies (1), there is a single path in the tree whose nodes satisfy the threshold, *i.e.* $p(y) > \tau \iff y \in \mathcal{A}(\hat{y})$. Inference can therefore be performed by traversing down from the root and the maximiser is unique assuming that $I(y)$ is strictly increasing on the edges of the tree. While this property is elegant, we also entertain arbitrary $p$ and $\tau < 0.5$ in the remainder of the paper. We refer to the special case of $\tau = 0.5$ as **majority** inference. Rather than consider a single threshold $\tau$, we propose to study the *operating characteristic curve* in the following section.

Two straightforward variants are **plurality** inference, which instead seeks the maximum-information label which is more likely than any other non-ancestor, and **information threshold** inference, which reverses the roles of the terms to instead maximise $p(y)$ subject to $I(y) \geq \zeta$. While the latter can also generate an operating curve, it does not provide adaptive specificity according to confidence.

Deng et al. [11] proposed an inference rule that maximises the expectation of a transformed reward

$$\xi_\lambda(p) = \arg\max_{y \in \mathcal{Y}} (I(y) + \lambda) p(y) \qquad (5)$$

with the factor $\lambda \geq 0$ chosen by bisection to yield the desired accuracy on a held-out set. Each value of $\lambda$ represents a particular balance between expected reward $I(y)p(y)$ and confidence $p(y)$. As $\lambda \to \infty$, the maximiser eventually becomes the root node, which maximises $p(y)$. Although $p(y)$ and $I(y)$ are monotonic with respect to depth (decreasing and increasing, respectively), there is no such guarantee for the product $I(y)p(y)$, and therefore an increase in $\lambda$ is not guaranteed to move the prediction towards the root. For $\lambda = 0$, we call this **expected information** inference.

Karthik et al. [20] and Deng et al. [9] proposed **conditional risk minimisation (CRM)** for leaf-node prediction. This scheme selects the leaf-node label $\hat{y}$ to minimise the expected cost $\mathbb{E}[C(Y, \hat{y})]$ with $Y \sim p(\cdot|x)$ being a random leaf node drawn from the predicted conditional likelihood

$$\xi(p) = \arg\min_{y \in \mathcal{L}} \sum_{u \in \mathcal{L}} C(u, y)\, p(u) \,. \qquad (6)$$

If $p$ satisfies (1), then CRM can be extended to predict non-leaf nodes by performing both the optimisation and the expectation over *all* nodes $\mathcal{Y}$ using the exclusive likelihood $\tilde{p}(y)$ of a node and *not* its children

$$\xi(p) = \arg\min_{y \in \mathcal{Y}} \sum_{u \in \mathcal{Y}} C(u, y)\, \tilde{p}(u) \,, \qquad \tilde{p}(u) = p(u) - \sum_{v \in \mathcal{C}(u)} p(v) \,. \qquad (7)$$

If we choose $C(y, \hat{y}) = -\text{Correct}(y, \hat{y})I(\hat{y})$ to maximise the expected reward, then CRM coincides with expected information inference when the ground-truth label is a leaf node. Different choices of $C(y, \hat{y})$ could be used to achieve different operating points, although it may be non-trivial to compute a complete trade-off curve with this approach.

## 4.2 Loss functions and likelihood parametrisations

We first present existing loss functions before introducing our novel loss functions. We use the notation that a vector $x \in \mathbb{R}^{\mathcal{U}}$ has elements $x_u \in \mathbb{R}$ indexed by $u \in \mathcal{U}$ and $x_{\mathcal{V}} \in \mathbb{R}^{\mathcal{V}}$ denotes the subvector on $\mathcal{V} \subseteq \mathcal{U}$. The sigmoid function is denoted $\sigma(x) = 1/(1 + \exp(-x))$ and scalar functions apply elementwise to vectors. The softmax function and cross-entropy loss operate on a given index set $\mathcal{U}$, defined $\text{softmax}_{\mathcal{U}} : \mathbb{R}^{\mathcal{U}} \to [0, 1]^{\mathcal{U}}$ and $\text{CE}_{\mathcal{U}} : \mathcal{U} \times \mathbb{R}^{\mathcal{U}} \to \mathbb{R}_+$ according to

$$[\text{softmax}_{\mathcal{U}}(\theta)]_u = \frac{\exp \theta_u}{\sum_{v \in \mathcal{U}} \exp \theta_v}, \quad \text{CE}_{\mathcal{U}}(u, \theta) = -\log [\text{softmax}_{\mathcal{U}}(\theta)]_u, \quad \text{for } u \in \mathcal{U}. \quad (8)$$

It will be useful to introduce the matrix $A \in \{0, 1\}^{\mathcal{Y} \times \mathcal{Y}}$ encoding the ancestor relation $A_{uv} = [u \supseteq v]$, such that the linear maps $x \mapsto Ax$ and $x \mapsto A^T x$ compute sums over descendants and ancestors, respectively. (The linear maps can be computed without explicitly instantiating the matrix.) Further, let $A_{\mathcal{L}} \in \{0, 1\}^{\mathcal{Y} \times \mathcal{L}}$ denote the leaf-node column subset, such that $x \mapsto A_{\mathcal{L}}x$ computes the sum over leaf descendants and $x \mapsto A_{\mathcal{L}}^T x$ computes the sum over the ancestors for each leaf node.

The most straightforward method is to use a **flat softmax** classifier with a parameter vector $\theta$ in $\mathbb{R}^{\mathcal{L}}$. The likelihoods of leaf nodes are obtained directly from the softmax, while the likelihoods of internal nodes are obtained by a recursive bottom-up sum:

$$q_y(\theta) = \begin{cases} [\text{softmax}_{\mathcal{L}}(\theta)]_y & y \in \mathcal{L} \\ \sum_{v \in \mathcal{C}(y)} q_v(\theta) & y \notin \mathcal{L}. \end{cases} \quad (9)$$

This can be succinctly expressed $q(\theta) = A_{\mathcal{L}} \text{softmax}_{\mathcal{L}}(\theta)$. The NLL reduces to the familiar (convex) cross-entropy $\ell(y, \theta) = \text{CE}_{\mathcal{L}}(y, \theta)$ for leaf-node labels $y \in \mathcal{L}$. However, it is non-convex for general labels $y \in \mathcal{Y}$, resembling a loss for Multiple Instance Learning [22]:

$$\ell(y, \theta) = -\log q_y(\theta) = -\log \sum_{u \in \mathcal{L}(y)} \exp \theta_u + \log \sum_{u \in \mathcal{L}} \exp \theta_u. \quad (10)$$

Bertinetto et al. [2] proposed an alternative loss for the same parametrisation, **Hierarchical Cross Entropy (HXE)**. This loss considers conditional distributions given the parent, placing geometrically decreasing weight on deeper nodes with discount factor $\gamma = e^{-\alpha} \in (0, 1]$. (For $\gamma < 1$, HXE is non-convex even for leaf-node labels $y \in \mathcal{L}$.) To define the loss, let $\omega_0, \ldots, \omega_{d(y)} \in \mathcal{Y}$ denote the ordered ancestors of $y$ (from $\omega_0 = \mathbf{r}$ to $\omega_{d(y)} = y$) in:

$$\ell_\alpha(y, \theta) = \sum_{k=1}^{d(y)} -\gamma^{k-1} \log \frac{q_{\omega_k}(\theta)}{q_{\omega_{k-1}}(\theta)} = \sum_{k=1}^{d(y)} \gamma^{k-1} \left[ -\log \sum_{u \in \mathcal{L}(\omega_k)} \exp \theta_u + \log \sum_{u \in \mathcal{L}(\omega_{k-1})} \exp \theta_u \right]. \quad (11)$$

Besides the flat softmax, another naïve baseline is the **multi-label sigmoid** classifier; that is, independent binary logistic regression per node. The likelihoods are parametrised by a vector $\theta$ in $\mathbb{R}^{\mathcal{Y} \setminus \{\mathbf{r}\}}$ according to $q(\theta) = \sigma(\theta)$. Clearly, this is not guaranteed to satisfy (1). Since the binary problems are imbalanced, we adopt the Focal Loss [23], which has two key hyper-parameters:

$$\ell_{\alpha,\gamma}(y, \theta) = \sum_{u \in \mathcal{Y}} \text{FL}_{\alpha,\gamma}([u \subseteq y], \sigma(\theta_u)), \quad \text{FL}_{\alpha,\gamma}(y, p) = \begin{cases} -\alpha(1-p)^\gamma \log p, & y = 1 \\ -(1-\alpha)p^\gamma \log(1-p), & y = 0. \end{cases}$$

Redmon and Farhadi [31] introduced a **conditional softmax** classifier, which uses a separate softmax for the conditional likelihood of child nodes given each parent. The parameter vector $\theta$ is in $\mathbb{R}^{\mathcal{Y} \setminus \{\mathbf{r}\}}$ and the likelihood of each node is obtained as a recursive top-down product. Let $r_y(\theta) = p(y|\pi(y), x)$ denote the conditional likelihood of $y$ given its parent to obtain:

$$r_{\mathcal{C}(y)}(\theta) = \text{softmax}_{\mathcal{C}(y)}(\theta_{\mathcal{C}(y)}), \quad q_y(\theta) = r_y(\theta) \cdot q_{\pi(y)}(\theta) = \prod_{u \in \mathcal{A}(y) \setminus \{\mathbf{r}\}} r_u(\theta). \quad (12)$$

This can be succinctly expressed $\log q(\theta) = A^T \log r(\theta)$. The conditional softmax has the same degrees of freedom as the flat softmax due to the invariance of the softmax function. The loss is taken to be NLL, which is a sum of cross-entropy losses and therefore convex for general labels $y \in \mathcal{Y}$:

$$\ell(y, \theta) = -\log q_y(\theta) = \sum_{u \in \mathcal{A}(y) \setminus \mathbf{r}} \text{CE}_{\mathcal{S}(u)}(u, \theta_{\mathcal{S}(u)}). \quad (13)$$

Brust and Denzler [4] similarly proposed a **conditional sigmoid** classifier for the more general problem of Multi-Path Labels in a DAG. We specialise this to a tree by replacing $r(\theta) = \sigma(\theta)$ above. This ensures that parents are more likely than each child, but does not guarantee the condition in (1). Their proposed loss function is Binary Cross Entropy for children of ancestors of the label:

$$\ell(y, \theta) = \sum_{u \in \mathcal{A}(y)} \sum_{v \in \mathcal{C}(u)} \text{BCE}([v \supseteq y], \sigma(\theta_v)), \quad \text{BCE}(y, p) = \begin{cases} -\log p, & y = 1 \\ -\log(1 - p), & y = 0. \end{cases} \quad (14)$$

Several works [32, 34, 39] have proposed to use *parameter sharing*, whereby the unnormalised score of each node is obtained as a sum of ancestor scores. That is, cumulative path scores $\beta \in \mathbb{R}^{\mathcal{Y}}$ are obtained from node scores $\theta$ in $\mathbb{R}^{\mathcal{Y}}$ according to $\beta_y = \sum_{u \in \mathcal{A}(y)} \theta_u$ or simply $\beta = A^T \theta$. In the context of deep learning, this approach was employed in the **Deep Realistic Taxonomic Classifier (Deep RTC)** of Wu et al. [39]. Motivated as a generalisation of "realistic classifiers", which may abstain from making a prediction [37], Deep RTC performs inference by greedy top-down traversal, using a threshold on unnormalised scores as a stopping condition. To map the unnormalised scores to $[0, 1]$, we apply the monotonic mapping $q(\theta) = \sigma(\beta)$. To ensure that internal nodes obtain high scores even when all ground-truth labels are leaf nodes, Deep RTC is trained using Stochastic Tree Sampling, taking the expected cross-entropy at the leaf nodes $\mathcal{K} \subset \mathcal{Y}$ of a random cut of the tree:

$$\ell(y, \theta) = \mathbb{E}_{\mathcal{K}} \left[ \text{CE}_{\mathcal{K}}(\text{proj}_{\mathcal{K}}(y), \beta_{\mathcal{K}}) \right], \quad \text{where } \{\text{proj}_{\mathcal{K}}(y)\} = \mathcal{A}(y) \cap \mathcal{K}. \quad (15)$$

We propose to consider an ablation of Deep RTC, the **Parameter Sharing (PS) softmax**, which is a simple linear reparametrisation of the flat softmax using the leaf-node scores from parameter sharing $\beta_{\mathcal{L}} = A_{\mathcal{L}}^T \theta$ with $\theta$ in $\mathbb{R}^{\mathcal{Y}}$. This can be succinctly expressed $q(\theta) = A_{\mathcal{L}} \text{softmax}_{\mathcal{L}}(A_{\mathcal{L}}^T \theta)$. Unlike Deep RTC, this yields likelihoods that satisfy (1).

Finally, we introduce two novel loss functions for a parametrisation that assigns non-zero mass to internal nodes. This parametrisation uses a softmax over *all* nodes $\mathcal{Y}$ with parameter vector $\theta$ in $\mathbb{R}^{\mathcal{Y}}$ to first obtain "exclusive" likelihoods $\tilde{q}_y(\theta)$ (*i.e.* likelihood of a node and *not* its children), and then obtains total likelihoods by a bottom-up recursive sum:

$$\tilde{q}(\theta) = \text{softmax}_{\mathcal{Y}}(\theta), \qquad q_y(\theta) = \tilde{q}_y(\theta) + \sum_{v \in \mathcal{C}(y)} q_v(\theta) \quad (16)$$

or, succinctly, $q(\theta) = A^T \text{softmax}_{\mathcal{Y}}(\theta)$. If we simply minimise the NLL, there would be little incentive to assign non-zero mass to internal nodes when most labels are leaf nodes (and the loss would be non-convex for non-leaf labels). We therefore propose the **soft-max-descendant** loss

$$\ell(y, \theta) = \sum_{u \in \mathcal{A}(y)} \frac{1}{|\mathcal{L}(u)|} \text{CE}_{\{u\} \cup \mathcal{N}(u)}(u, \theta) \quad (17)$$

where $\mathcal{N}(y) = \mathcal{Y} \setminus (\mathcal{A}(y) \cup \mathcal{D}(y))$ is the set of incorrect (negative) labels for $y$; that is, labels which are neither ancestors nor descendants. This loss aims to ensure that each ancestor of the ground-truth label is classified positively against all incorrect labels. We call it soft-max-descendant because it takes the log-sum-exp (like a soft maximum) over incorrect sub-trees. Normalisation by the number of leaf descendants is necessary to prevent higher-level nodes from dominating the loss. It will later be seen that this loss is ineffective; it is included as a negative result.

Finally, inspired by recent work using logit adjustment for long-tail learning [25], we consider a **soft-max-margin** loss [29, 15, 13], which is a soft version of the structured hinge loss [35], defined

$$\ell(y, \theta) = \text{CE}_{\mathcal{Y}}(y, \theta + \alpha C(y, \cdot)) = \log \left( 1 + \sum_{y' \in \mathcal{Y} \setminus \{y\}} \exp\{\theta_{y'} - \theta_y + \alpha C(y, y')\} \right) \quad (18)$$

where $C(y, y')$ gives the desired score margin between $y$ and $y'$. For hierarchical classification, we adopt $C(y, y') = 1 - \text{Correct}(y, y')$ to seek greater separation of the ground-truth label from incorrect classes than from correct classes, and we found $\alpha = 5$ to provide the best results. Note that the logit adjustment is only employed during training. Despite the original "hard" structured hinge loss also being convex, we were unable to use it to train a deep model to high accuracy.

## 5 Operating characteristic curve

Given a classifier that predicts a conditional distribution over the class hierarchy, the set of predictions that can be obtained using confidence-threshold inference with *some* value of $\tau$ is the Pareto set; that is, the set of classes such that no other class is both more confident and more informative:

$$\mathcal{H}(p(\cdot|x)) = \{y \in \mathcal{Y} : \nexists u(p(u|x) > p(y|x) \wedge I(u) > I(y))\}. \quad (19)$$

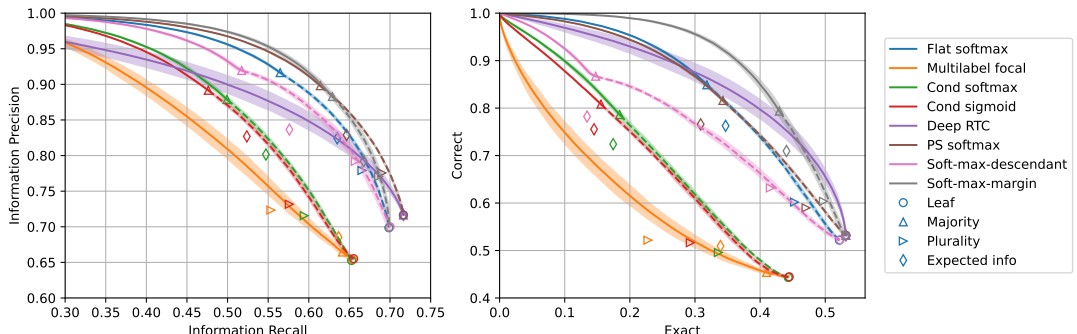

Figure 1: Correctness-specificity trade-offs for different methods and metrics on the iNat21 validation set. Markers indicates operating points obtained by inference functions. Comparing methods using single operating points would not provide a complete perspective. The dashed line is the part of the curve with threshold below 0.5. The shaded areas depict two standard deviations to either side.

For a single example $(x, y)$, let us define the sequence $\hat{y}_0, \ldots, \hat{y}_{K-1}$ to denote the elements of the Pareto set, ordered such that $p(\hat{y}_k|x) \geq p(\hat{y}_{k+1}|x)$ and $I(\hat{y}_k) \leq I(\hat{y}_{k+1})$. Further let $s_k = p(\hat{y}_k|x)$ denote the corresponding confidence score such that the prediction is a piecewise-continuous function of $\tau$ with $\hat{y}(\tau) = \hat{y}_k$ for $s_k > \tau \geq s_{k+1}$. The value of any metric $M(y, \hat{y}(\tau))$ is also a piecewise-continuous function of $\tau$, taking values $z_k = M(y, \hat{y}_k)$.

The cumulative metric $Z(\tau)$ for a set of examples $\{(x^i, y^i)\}_i$ is a sum of piecewise-constant functions $M(y^i, \hat{y}^i(\tau))$ and therefore piecewise-constant itself, with the form $Z(\tau) = Z_j$ for $S_j > \tau \geq S_{j+1}$. The values of $Z_j$ and $S_j$ can be obtained by introducing $\delta_k^i = z_k^i - z_{k-1}^i$ and reordering sums:

$$Z(\tau) = \sum_i M(y^i, \hat{y}^i(\tau)) = \sum_i (z_0^i + \sum_{k:s_k^i > \tau} \delta_k^i) = \sum_i z_0^i + \sum_{(i,k):s_k^i > \tau} \delta_k^i. \qquad (20)$$

Therefore $Z_j$ is obtained as a partial sum $Z_j = Z_0 + \sum_{u=1}^{j} \Delta_u$ with $Z_0 = \sum_i z_0^i$ and the sequences $S_j$ and $\Delta_j$ obtained by merging the ordered lists of $(s_k^i, \delta_k^i)$ pairs to be descending in $s$. The following section will compare different methods using parametric curves of correctness and specificity metrics as a function of $\tau$.

The ordered Pareto set for a single example can be obtained in $O(|\mathcal{Y}| \log |\mathcal{Y}|)$ time. For arbitrary $p$ and $I$, the worst-case running time for a dataset of size $N$ is $O(N|\mathcal{Y}|(\log N + \log |\mathcal{Y}|))$, although each Pareto set can be computed independently in parallel. If we assume that the tree is relatively balanced, $p(\cdot|x)$ satisfies (1) and restrict ourselves to $\tau \in [0.5, 1]$, then the size of the set is bounded by the maximum depth of the tree, which is $O(\log |\mathcal{Y}|)$. This results in a running time of $O(N|\mathcal{Y}| \log |\mathcal{Y}|)$ to find the Pareto sets and $O(N \log N \log |\mathcal{Y}|)$ to merge the sequences. Otherwise, ensuring that all leaf nodes have equal information guarantees that the Pareto set contains at most one leaf node, reducing the worst-case merge time by a factor of $1 - |\mathcal{L}|/|\mathcal{Y}|$.

## 6 Empirical study

We now apply this technique to compare different loss functions. Most experiments consider image classification on the iNat21-Mini dataset [36], containing 50 examples each for 10,000 biological species in a seven-level taxonomy. All examples have leaf-node (full-depth) annotation. For all iNat21 experiments, we use a ResNet-18 model [19] with input images of size 224×224. We start from the Pytorch ImageNet-pretrained checkpoint [28] and train for 20 epochs using SGD with momentum 0.9, cosine schedule [24], batch size 64, initial learning rate 0.01 and weight decay 0.0003. Most experiments were conducted on a single machine with one Nvidia A6000 GPU. Each epoch of iNat21-Mini takes about 20 minutes (5-8 minibatches per second). To obtain error-bars, we used a larger, shared machine with 16 Nvidia V100 GPUs (still using one GPU to train each model). The code was implemented using the Pytorch library [28] and is available under the MIT license.

Table 1: Overall metrics derived from operating curves on the iNat21 validation set. Average Precision (AP) and Average Correct (AC) are integrals with respect to Recall, while R@$X$C indicates Recall at $X$% Correct. All precision and recall metrics use information as the specificity measure (not depth). The error ranges are two standard deviations. *Deep RTC and PS softmax actually use the same model weights with different output parametrisations since the optimal cut probability for Deep RTC was found to be zero.

| | From operating curves | | | | Majority | Leaf |
| % | AP | AC | R@90C | R@95C | $F_1$ | $F_1$ |
|---|---|---|---|---|---|---|
| Flat softmax | 66.90 ±0.16 | 64.71 ±0.18 | 52.44 ±0.42 | 45.88 ±0.30 | 63.68 ±0.19 | 69.86 ±0.13 |
| Multilabel focal | 58.75 ±0.24 | 54.08 ±0.20 | 32.64 ±0.16 | 26.80 ±0.28 | 65.05 ±0.24 | 65.38 ±0.29 |
| Cond softmax [31] | 60.72 ±0.11 | 57.47 ±0.13 | 42.03 ±0.12 | 35.71 ±0.20 | 58.54 ±0.20 | 65.27 ±0.11 |
| Cond sigmoid [4] | 60.64 ±0.17 | 57.13 ±0.16 | 40.52 ±0.13 | 34.29 ±0.18 | 56.73 ±0.07 | 65.52 ±0.19 |
| Deep RTC [39]* | 66.05 ±0.16 | 60.67 ±0.15 | 30.56 ±0.79 | 17.64 ±1.01 | **71.65** ±0.12 | **71.65** ±0.12 |
| PS softmax* | **68.90** ±0.13 | **66.86** ±0.12 | *55.55* ±0.22 | *49.13* ±0.15 | 67.97 ±0.19 | *71.65* ±0.12 |
| Soft-max-descendant | 66.46 ±0.37 | 64.19 ±0.39 | 49.15 ±0.36 | 43.29 ±0.56 | 60.99 ±0.33 | 70.01 ±0.33 |
| Soft-max-margin | *67.63* ±0.25 | *65.97* ±0.26 | **56.53** ±0.28 | **50.65** ±0.41 | 67.92 ±0.27 | 69.89 ±0.24 |

## 6.1 Operating characteristic curves

Our main experiment trains and evaluates each of the methods on the iNat21-Mini dataset to obtain operating curves. The results presented here are $\mu \pm 2\sigma$ from five trials with different random seeds.

Figure 1 presents curves for precision-vs-recall and correct-vs-exact. The most striking observation is that all methods except PS softmax, Deep RTC [39] and our soft-max-margin loss are virtually dominated by the flat softmax classifier at all operating points. For the precision-recall trade-off, PS softmax is the best at high recall and soft-max-margin is the best at high precision. The soft-max-margin has a distinct advantage in the correct-exact trade-off and Deep RTC is more competitive. The least effective method is the binary multilabel baseline, followed by the two approaches that learn top-down conditional distributions [31, 4]. The singular operating points achieved by different inference methods are also depicted in the figure, with the operating curve always containing majority inference and leaf inference as special cases. For parametrisations $p(\cdot|x) = q(\theta)$ that respect the hierarchy, specificity and correctness should be monotonic in $\tau$ for $\tau > 0.5$, hence a dashed line is used for the segment with $\tau < 0.5$. It is observed that plurality inference enables more-specific predictions than majority inference. However, both the plurality and the expected information methods lie below the curve obtained by confidence threshold inference. Further enquiry is required to understand the cause of the "knot" observed at $\tau = 0.5$ in the curves for the soft-max-descendant method. Critically, the curves are much more useful than single operating points for the purposes of algorithm development and selection.

Table 1 further presents several integral and intercept metrics obtained from the curves. For comparison, the $F_1$ metrics for majority and leaf inference are shown. PS softmax and soft-max-margin achieve the best results across all metrics.

Several of the proposed loss functions have additional hyper-parameters to specify. Figure 4 in the appendix examines the impact of label smoothing with flat softmax, focal loss parameters in the multilabel classifier, discount factor in HXE [2] and cut probability in Deep RTC [39]. Label smoothing and HXE include the flat softmax as a special case, and this was the optimal hyper-parameter choice in both cases. For this reason, they were excluded from our main experiment. The optimal cut probability for Deep RTC was zero, meaning it reduces to a softmax loss with logits obtained by parameter sharing. This was surprising, as it provides no incentive for the model to increase the scores of internal nodes.

We conducted additional experiments with a subset of loss functions on ImageNet-1k using the hierarchy of [2], training a ResNet-50 model from scratch. Figure 2 presents operating curves for this experiment. The observations are mostly consistent: our soft-max-margin method achieves a better trade-off than existing methods, Deep RTC is strong at high recall (specificity) but mediocre at high precision (correctness), and the conditional softmax is worse than the flat softmax across the full range. Unlike the previous dataset, PS softmax underperformed the flat softmax. This experiment similarly used batch size 64, weight decay 0.0003 and initial learning rate 0.01, but was only trained for 15 epochs. Figure 2 also shows the impact of using each loss function to train a *linear* model on top of the feature representation of a CLIP [30] model with ViT-B/32 [12] architecture. The

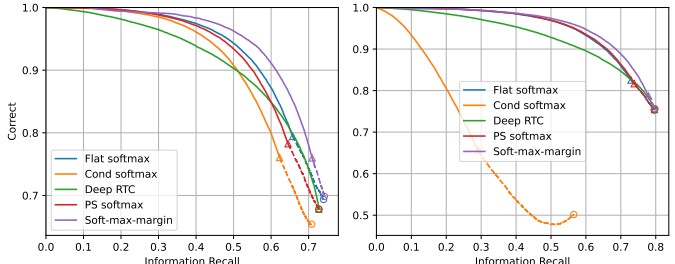

Table 2: iNat21-Mini accuracy at level $j$ when training at level $i$.

| % | $j = 3$ | 4 | 5 | 6 |
|---|---|---|---|---|
| $i = 3$ | 82.7 | | | |
| 4 | 85.2 | 60.7 | | |
| 5 | 86.6 | 64.6 | 50.6 | |
| 6 | 86.9 | 68.2 | 56.8 | 44.6 |
| 7 | **87.3** | **70.4** | **59.8** | **49.2** |

Figure 2: ImageNet-1k operating curves for (left) ResNet-50 from scratch and (right) linear model with CLIP ViT-B/32 features.

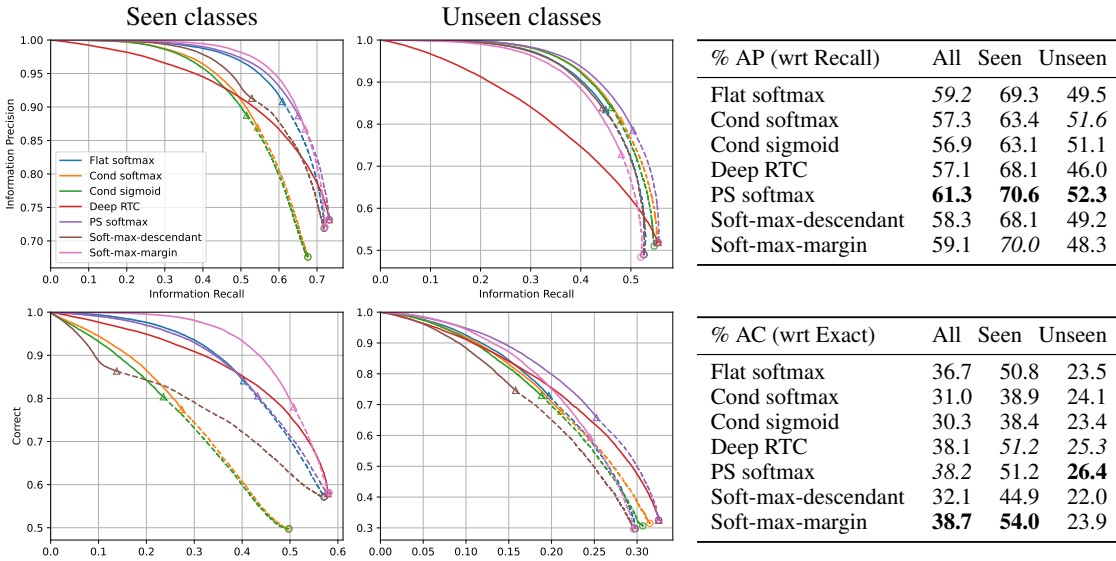

| % AP (wrt Recall) | All | Seen | Unseen |
|---|---|---|---|
| Flat softmax | *59.2* | 69.3 | 49.5 |
| Cond softmax | 57.3 | 63.4 | *51.6* |
| Cond sigmoid | 56.9 | 63.1 | 51.1 |
| Deep RTC | 57.1 | 68.1 | 46.0 |
| PS softmax | **61.3** | **70.6** | **52.3** |
| Soft-max-descendant | 58.3 | 68.1 | 49.2 |
| Soft-max-margin | 59.1 | *70.0* | 48.3 |

| % AC (wrt Exact) | All | Seen | Unseen |
|---|---|---|---|
| Flat softmax | 36.7 | 50.8 | 23.5 |
| Cond softmax | 31.0 | 38.9 | 24.1 |
| Cond sigmoid | 30.3 | 38.4 | 23.4 |
| Deep RTC | 38.1 | *51.2* | *25.3* |
| PS softmax | *38.2* | 51.2 | **26.4** |
| Soft-max-descendant | 32.1 | 44.9 | 22.0 |
| Soft-max-margin | **38.7** | **54.0** | 23.9 |

Figure 3: Operating curves for the out-of-distribution experiment on iNat21, divided into examples from seen and unseen classes. The top-down methods (conditional softmax, conditional sigmoid) are much more competitive on unseen classes. The soft-max-margin is most effective on seen classes but PS softmax is the most robust.

soft-max-margin still provides an improvement over the baseline, while the PS softmax model is mathematically equivalent to a flat softmax (except for weight decay). Compared to training from scratch, higher recall is achieved, presumably due the strong feature representation. However, the conditional softmax is much worse, suggesting that the high-level classes are less easily separated in this representation. The linear model was trained for 20 epochs with batch size 256, initial learning rate 0.1 and weight decay $10^{-5}$.

## 6.2 Flat classifiers at different levels

It was unexpected that the conditional softmax would fail to out-perform the flat softmax at *any* operating point. One possible explanation is that the high-level classes are better learned using a union of low-level classifiers than a single high-level classifier because the latter effectively ignores the fine-grained annotations. To investigate this hypothesis, we trained flat models at each level of the hierarchy and compared the classifiers trained at level $i$ with those trained at levels $i + 1$, $i + 2$ and so on, shown in Table 2. Classifiers trained at deeper levels consistently achieved higher accuracy, although with diminishing returns. This motivated the design of our proposed loss functions, which predict scores for low-level and high-level classes.

### 6.3 Unseen classes

We hypothesised that the parametrisations which explicitly estimate scores for internal nodes might achieve better generalisation to examples from "unseen" classes, which are not present during training but still belong to an internal node. To evaluate the ability of models to correctly classify such examples, we randomly select half of the leaf nodes and remove them from the dataset *and* hierarchy during training. At test time, examples from these classes were retained with their labels projected onto the sub-tree.

Figure 3 presents the operating curves and integral metrics for the examples from seen and unseen classes. The conditional softmax [31] and conditional sigmoid [4] are much more competitive for unseen classes, although PS softmax remains the most effective. The soft-max-margin approach is excellent for seen classes and relatively poor for unseen classes, suggesting that it learns a highly tailored model that could in fact be worse than the naïve baseline in an out-of-distribution setting. The soft-max-descendant approach is uncompetitive in both settings.

## 7 Conclusion

This work has presented a new approach for evaluating hierarchical classifiers for the prediction of non-leaf nodes. It was shown that the flat softmax classifier, despite having no knowledge of the hierarchy during training, is nonetheless a highly competitive baseline. Novel loss functions were proposed, and the soft-max-margin loss was shown to out-perform several existing methods. On the other hand, the proposed soft-max-descendant losses were shown to be ineffective. Surprisingly, top-down classifiers were found to be inferior across the entire operating range. Evidence suggests that this is due to the difficulty of learning a classifier to separate high-level classes without fine-grained supervision. The top-down classifiers were shown to be more effective on a synthetic out-of-distribution experiment, where recognition of super-classes is required.

One potential disadvantage of the soft-max-margin loss is that it requires manual design of the margin. Future work could investigate different margins, direct optimisation of the area under the curve [3] or the impact of calibration [17]. It would also be interesting to generalise parameter sharing to DAGs, to evaluate hierarchical loss functions in a long-tail setting, and to investigate other problems where the soft-max-margin loss could be applied. One notable limitation of using a fully-connected prediction layer is that it may not scale to millions of classes, where embeddings or metric learning are more suitable.

**Social impact.** While hierarchical classifiers may mitigate some of the risks of misclassification by their ability to fail gracefully, this may enable more widespread use of automatic classification or lead human users to place more trust in the system. Caution should always be exercised when employing automatic classification in consequential settings. A class hierarchy with *tree* structure may also encourage an oversimplified view of a problem due to its inability of different classes to have a non-empty intersection.

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
