# Hierarchical classification
# at multiple operating points

**Jack Valmadre**
Australian Institute for Machine Learning, University of Adelaide
`jack.valmadre@adelaide.edu.au`

## A Impact of hyper-parameters

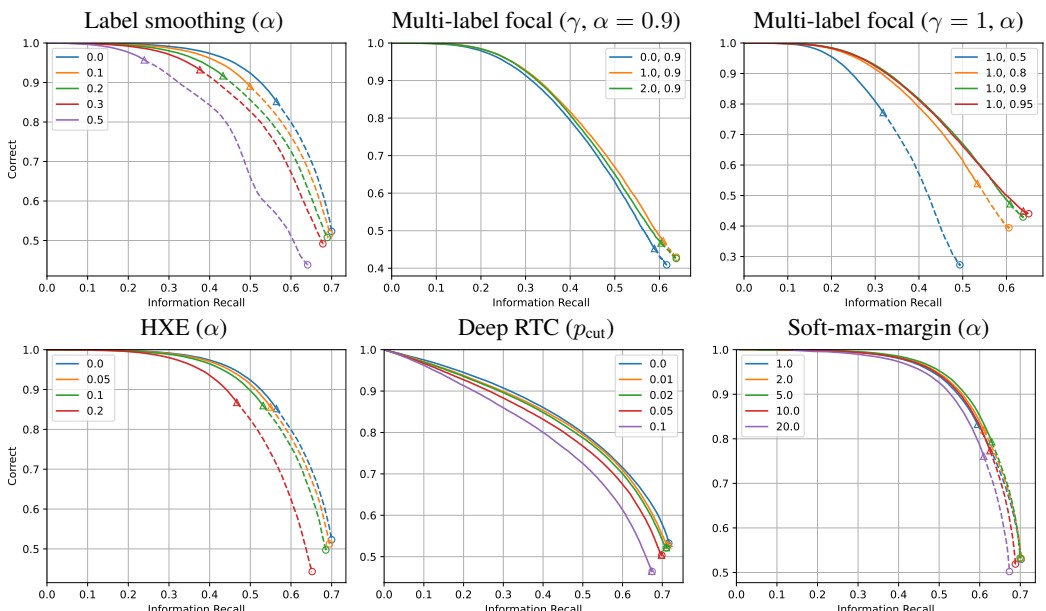

Figure 4: Impact of loss hyper-parameters on trade-off with iNat21-Mini (correct vs. recall). Label smoothing and HXE achieve their best accuracy when set to zero, which is equivalent to a flat softmax. The soft-max-margin loss with $C(y, \hat{y}) = 1 - \text{Correct}(y, \hat{y})$ performs best using scaling factor $\alpha \approx 5$.

36th Conference on Neural Information Processing Systems (NeurIPS 2022).

# B  Table of parametrisations

Table 3 outlines the parametrisation that corresponds to each loss function. The loss functions that use a sigmoid do not guarantee a valid distribution on the class hierarchy (eq. 1). Note that we use confidence threshold inference for all loss functions, regardless of the inference function that was used in the original publication.

Table 3: Definition and properties of the parametrisations used by each loss function.

| Loss | $\theta \in$ | Parametrisation | Properties |
|---|---|---|---|
| Flat softmax, HXE [2] | $\mathbb{R}^{\mathcal{L}}$ | $p(y) = \sum_{v \in \mathcal{L}(y)} \mathrm{softmax}_v(\theta)$ | $p(y) \in [0,1]$ 
 $p(y) \geq \sum_{v \in \mathcal{C}(y)} p(v)$ 
 $\sum_{y \in \mathcal{L}} p(y) = 1$ |
| Multi-label | $\mathbb{R}^{\mathcal{Y} \backslash \{\mathbf{r}\}}$ | $p(y) = \sigma(\theta_y)$ | $p(y) \in [0,1]$ |
| Conditional softmax [31] | $\mathbb{R}^{\mathcal{Y} \backslash \{\mathbf{r}\}}$ | $p(y) = \prod_{u \in \mathcal{A}(y) \backslash \{\mathbf{r}\}} p(u\|\pi(u)), \quad p(y\|\pi(y)) = \mathrm{softmax}_y(\theta_{\mathcal{C}(\pi(y))})$ | $p(y) \in [0,1]$ 
 $p(y) \geq \sum_{v \in \mathcal{C}(y)} p(v)$ 
 $\sum_{y \in \mathcal{L}} p(y) = 1$ |
| Conditional sigmoid [4] | $\mathbb{R}^{\mathcal{Y} \backslash \{\mathbf{r}\}}$ | $p(y) = \prod_{u \in \mathcal{A}(y) \backslash \{\mathbf{r}\}} p(u\|\pi(u)), \quad p(y\|\pi(y)) = \sigma(\theta_y)$ | $p(y) \in [0,1]$ 
 $p(y) \geq p(v), \quad v \in \mathcal{C}(y)$ |
| Deep RTC [39] (training; random cut $\mathcal{K}$) | $\mathbb{R}^{\mathcal{Y}}$ | $p_{\mathcal{K}}(y) = \mathrm{softmax}_y(\beta_{\mathcal{K}}), \qquad \beta_y = \sum_{u \in \mathcal{A}(y)} \theta_u$ | $p_{\mathcal{K}}(y) \in [0,1]$ 
 $\sum_{y \in \mathcal{K}} p_{\mathcal{K}}(y) = 1$ |
| Deep RTC [39] (inference) | $\mathbb{R}^{\mathcal{Y}}$ | $p(y) = \sigma(\beta_y), \qquad \beta_y = \sum_{u \in \mathcal{A}(y)} \theta_u$ | $p(y) \in [0,1]$ |
| PS softmax | $\mathbb{R}^{\mathcal{Y}}$ | $p(y) = \sum_{v \in \mathcal{L}(y)} \mathrm{softmax}_v(\beta_{\mathcal{L}}), \; \beta_y = \sum_{u \in \mathcal{A}(y)} \theta_u$ | $p(y) \in [0,1]$ 
 $p(y) \geq \sum_{v \in \mathcal{C}(y)} p(v)$ 
 $\sum_{y \in \mathcal{L}} p(y) = 1$ |
| Soft-max-descendant, Soft-max-margin | $\mathbb{R}^{\mathcal{Y} \backslash \{\mathbf{r}\}}$ | $p(y) = \sum_{v \in \mathcal{D}(y)} \mathrm{softmax}_v(\theta)$ | $p(y) \in [0,1]$ 
 $p(y) \geq \sum_{v \in \mathcal{C}(y)} p(v)$ |

# C Algorithms

**Algorithm 1** Algorithm for finding ordered Pareto set. (Algorithm 3.1 from Kung et al. (1975), modified to admit non-unique values.) The inputs $x$ and $y$ are lists with equal length. The sorting algorithm must be stable such that successive sorts yield lexicographic order.

> **procedure** ORDEREDPARETOSET($x, y$)
>   $\pi \leftarrow \text{argsort}(-y)$                               $\triangleright$ sort descending by $y$
>   $x, y \leftarrow x[\pi], y[\pi]$
>   $\rho \leftarrow \text{argsort}(-x)$                               $\triangleright$ sort descending by $x$
>   $x, y, \pi \leftarrow x[\rho], y[\rho], \pi[\rho]$     $\triangleright$ preceding pairs have greater $x$ (or greater $y$ if $x$ are equal)
>   $T \leftarrow \text{cummax}(y)$                         $\triangleright$ get maximum $y$ over preceding pairs
>   $\omega \leftarrow [i : i = 0 \vee T[i-1] < y[i]]$       $\triangleright$ take subset where $y$ exceeds all preceding pairs
>   $x, y, \pi \leftarrow x[\omega], y[\omega], \pi[\omega]$     $\triangleright$ have monotonicity $x[i] \geq x[i+1]$ and $y[i] \leq y[i+1]$
>   **return** $\pi$                                      $\triangleright$ return indices in original list
> **end procedure**

**Algorithm 2** Construct trade-off curve for dataset of $n$ examples. Assume that the nodes of the tree are represented by indices $\mathcal{Y} = \{0, 1, \ldots, |\mathcal{Y}| - 1\}$. Each $y^i$ is a label in $\mathcal{Y}$ and each $p^i$ is a list of length $|\mathcal{Y}|$. We use square brackets to denote array elements (subscripts were used in the main text).

> **procedure** CONSTRUCTDATASETCURVE($\{y^i, p^i\}_{i=1}^n, I, \tau_{\min}$)
>   $Z_0 \leftarrow 0$
>   $S \leftarrow []$
>   $\Delta \leftarrow []$
>   **for** $i = 1, \ldots, n$ **do**
>     $\omega \leftarrow [y \in \mathcal{Y} : p^i[y] > \tau_{\min}]$        $\triangleright$ select subset that satisfy minimum threshold
>     $\pi \leftarrow \text{OrderedParetoSet}(p^i[\omega], I[\omega])$        $\triangleright$ subset of nodes that could be predicted
>     $\hat{y} \leftarrow \omega[\pi]$
>     $K \leftarrow \text{len}(\hat{y})$
>     $z[k] \leftarrow M(y^i, \hat{y}[k]), \quad 0 \leq k < K$        $\triangleright$ evaluate metric for all predictions
>     $\delta[k] \leftarrow z[k] - z[k-1], \quad 1 \leq k < K$        $\triangleright$ get relative change in metric
>     $s[k] \leftarrow p^i[\hat{y}[k]], \quad 1 \leq k < K$        $\triangleright$ get threshold for change
>     $Z_0 \leftarrow Z_0 + z[0]$
>     $S \leftarrow \text{concat}(S, s)$
>     $\Delta \leftarrow \text{concat}(\Delta, \delta)$
>   **end for**
>   $\pi \leftarrow \text{argsort}(-S)$
>   $S, \Delta \leftarrow S[\pi], \Delta[\pi]$
>   $Z \leftarrow (1/n)(Z_0 + \text{cumsum}(\Delta))$        $\triangleright$ total metric is initial plus sum of changes
>   **return** $Z, S$
> **end procedure**