# OpenReview forum: "Hierarchical classification at multiple operating points"
_NeurIPS.cc/2022/Conference — NeurIPS 2022 Accept_

### Official Review · Reviewer_53HE · 2022-07-10

**Rating:** 6
**Confidence:** 3
**Soundness:** 3 good
**Presentation:** 2 fair
**Contribution:** 3 good

**Summary:**

Topic: the paper reviews the performance of hierarchical classification for deep learning classifiers.

They propose as scoring for internal nodes to sum up the softmax components of the leaves.
They propose for the above scoring two loss functions.
They propose in section 5 a way to compute curves over varying thresholds \tau for node probabilities p(y), where the graphs show correct versus exact and precision vs recall, with the note that these measures are defined in the sense of hierarchical measures and not as conventional precision vs recall for flat classifiers.

They evaluate these curves for a number of methods.
They run an experiment with simulated unseen classes which are part of the hierarchy.

The topic is relevant.
For a better score, section 5 needs to be verifiable and better understandable.


**Questions:**

Questions:

Are the p(y) for all methods probabilities? if yes, what is the set over which they sum up? That can be put in a table.


How does computing the curves in Section 5 work?
The ordering is a double ordering, with two criteria. How to sort with two criteria?
What is varied in line 233 with z_k and the \tau when considering the z_k? There is an unclear y in the definition of z_k which might be related to the \tau.
How to merge the sequence of pairs \delta into \Delta?

What is the meaning of eq(7) given the specific cost function choice C?
What is the meaning of eq(17) given the specific cost function choice C? It seems to give a higher weight to nodes y' such that the prediction y is on the path from y' ?




**Limitations:**

Some discussion has been made. No issue regarding potential negative societal impact.

**Strengths And Weaknesses:**


Strengths:
A certain novelty, with combinations of loss functions and section 5, if correct.
It is an overview evaluation for a number of methods.
The topic is interesting and relevant.

Weaknesses:

It has a not so great readability.
Some parts appear to be too brief, and it appears as if it would profit from an extended journal version.

-the meaning of the choice of C(y,\hat{y}) in line 181 is a bit unclear. One seems to sum I(y) values from ancestor nodes starting at the prediction \hat{y}.
-eq (7) needs a discussion
-the changes made to deepRTC need a discussion.
-eq (17) needs a discussion with respect to the concrete C term used.

-Section 5 is hard to understand.

The ordering is a double ordering, with two criteria. How to sort with two criteria?

p(\hat{y}_{k}) \ge p(\hat{y}_{k+1}) and I(\hat{y}_{k}) \le I(\hat{y}_{k+1}). What to do if p(\hat{y}_{k}) \ge p(\hat{y}_{k+1}) and I(\hat{y}_{k}) \ge I(\hat{y}_{k+1}) for example ?

There is a \Delta seemingly out of nowhere. How to merge the sequence of pairs ? Why does it work?
There is an unclear y in the definition of z_k which might be related to the \tau.

Section 5 reads as if one has to trust that it works as described.

-a smaller issue: it could be worth to cite some pre-deep learning hierarchical papers (but that causes no reduction in scoring) like:

Blaschko, Zaremba, Gretton Taxonomic Prediction with Tree-Structured Covariances
Zweig, Weinshall, Exploiting Object Hierarchy: Combining Models from Different Category Levels
Marszalek, Schmid, Semantic Hierarchies for Visual Object Recognition
and the like

---

> ### Author Response · Authors · 2022-08-01
> **Primary response to Reviewer 53HE**
>
> > Strengths: A certain novelty [in] loss functions and section 5, if correct
>
> Thanks for considering the manuscript in detail and recognizing these aspects.
> We believe the paper, especially Section 5, can be significantly improved given the feedback.
>
> > Are the p(y) for all methods probabilities? … That can be put in a table.
>
> Not all parameterizations give valid probabilities on the hierarchy: some satisfy eq. 1, some only guarantee that each child is less confident than its parent, and some do not even provide this.
> A table is a great idea.
> We can either move the equations of Section 4.2 into this table to make space, or add it as an appendix.
>
> > If [the p(y) are probabilities], what is the set over which they sum up?
>
> Indeed, the probabilities of all classes will not sum to 1 since some classes are supersets of others (think: event space not sample space).
> We instead say that a distribution is valid if p(y) ≥ 0 for all y, p(root) = 1 and p satisfies eq. 1.
>
> In the case where eq. 1 holds with *equality*, the likelihoods of the leaf nodes sum to 1.
> However, we deliberately permit superclasses to be "larger" than the union of their children, as this may help generalize to unseen classes.
> Still, if p is valid, the "exclusive" probabilities (i.e. of a node and _not_ its children), \\(\\tilde{p}(u)=p(u)-\\sum\_{v\\in \\mathcal{C}(u)}p(v)\\), will sum to 1 over all nodes.
>
> > How does computing the curves in Section 5 work?
>
> Each aspect is addressed below.
>
> > How to sort with two criteria?
>
> The key here is that we find an ordering of _just_ the Pareto set (not the set of all nodes), and the Pareto set (for 2D vectors) can always be ordered such that one criterion is increasing and the other is decreasing.
> (Visually, imagine walking along the Pareto front: https://upload.wikimedia.org/wikipedia/commons/b/b7/Front_pareto.svg)
> This is not self-evident and we will make it explicit.
>
> Another way to understand this is: the inference rule (eq. 4) will only predict a node with *lower confidence* if it contains *greater information*.
>
> In the example provided, if \\(p(\\hat{y}\_k)\\ge p(\\hat{y}\_{k+1})\\) and \\(I(\\hat{y}\_k)\\ge I(\\hat{y}\_{k+1})\\), then \\(\\hat{y}\_{k+1}\\) would not be in the Pareto set from eq. 18, as it is dominated by \\(\\hat{y}\_k\\).
>
> > What is varied in line 233 with z\_k and the τ when considering the z\_k? There is an unclear y in the definition of z\_k which might be related to the τ.
>
> At this point, we are considering a single example (x, y), so y is the GT label and is not related to τ.
> The sequence z\_k represents the value of the metric \\(M(y,\\hat{y})\\) for the sequence of predictions \\(\hat{y}\_k\\), hence the dependence on the GT label y.
>
> As τ is varied (decreased), the predicted label will progress through the sequence \\(\hat{y}\_k\\), with the predictions becoming less confident and more informative.
>
> Maybe it's more clear if we define the prediction as a piecewise-constant function \\(\\hat{y}(τ)=\\hat{y}\_{κ(τ)}\\) where \\(κ(τ)\\) finds the index k such that \\(c\_k>τ\\ge c\_{k+1}\\)?
> Then \\(M(y,\\hat{y}(τ)) = M(y,\\hat{y}\_{κ(τ)})=z\_{κ(τ)}\\).
>
> > How to merge the sequence of pairs δ into Δ? Why does it work?
>
> The lists of (confidence, delta\_metric) pairs for every example are merged to be ordered (descending) by confidence.
>
> The algorithm works by a simple manipulation of the summations over (i) examples and (ii) deltas.
> We propose to show this by adding an equation: (here, i is the index over examples)
> $$\\textstyle\\sum_i M(y^i,\\hat{y}^i(τ))=\\sum_i (z^i\_0+\\sum\_{k:c^i\_k>τ}δ^i\_k)=\\sum\_i z^i\_0+\\sum\_{(i,k):c^i\_k>τ}δ^i\_k=Z\_0+\\sum\_{j:C\_{j}>τ}\\Delta\_j$$
>
> Thus the metric for all τ can be obtained using a partial sum over \\(\\{δ^{i}\_{k}\\}_{i, k}\\) sorted by \\(c^i\_k\\).
>
> > Section 5 is hard to understand [and] reads as if one has to trust that it works.
>
> Besides the above changes, we will add algorithms to the appendix as suggested by reviewer @CWNt.
> The code release will include unit tests for these subroutines.
>
> > What is the meaning of eq(17) given the specific cost function choice C? It seems to give a higher weight to nodes y' such that the prediction y is on the path from y'?
>
> Sorry, there was an error here: it should be \\(C(y,y')=1-\\text{Correct}(y,y')\\).
> It's not exactly a weighting; rather, the meaning of the loss is to demand greater separation (in logit values) of the true label from the incorrect classes than from the other correct classes.
> This will be added to the text.
>
> > What is the meaning of eq(7) given the specific cost function choice C?
>
> Sorry, there was an error here: it should be \\(C(y,y')=-\\text{Correct}(y,y')I(y')\\).
> This seeks to predict the label y' that maximizes the expected correct information.
>
> > changes made to deepRTC need a discussion
>
> DeepRTC-softmax first obtains leaf likelihoods and adds these up (like flat softmax) to ensure a valid distribution.
> This will be more clear using the table of methods.

---

> > ### Comment · Reviewer_53HE · 2022-08-10
> > **thank you**
> >
> > This helps with the understanding

---

### Official Review · Reviewer_Kenf · 2022-07-10

**Rating:** 6
**Confidence:** 3
**Soundness:** 3 good
**Presentation:** 2 fair
**Contribution:** 3 good

**Summary:**

The authors consider the tasks of hierarchical classification where predictions can be done at different levels of granularity, which can be seen in datasets like ImageNet, where a super class can be spiders and its fine-grained classes are the different types of spiders.

The authors describe and compare different existing loss functions on a new metric class based on "Operating characteristic curve" on the  iNat21-Mini dataset. They also propose two loss functions: soft-max-descendant that scores internal nodes in the classification hierarchy, and soft-max-margin which is a soft version of the structured hinge loss.

Discussions have been made about how different loss functions behave based on the structure of the problem such as out-of-distribution and in-distribution.

**Questions:**

Can the authors explain the main contributions and motivation of this work? It is not clear why having "Operating characteristic curve" is important and why having the two loss functions "soft-max-descendant" and "soft-max-margin" are important as their performance is poor in many settings.

A lot of loss functions have been compared against a new metric, what is the takeaway? What loss function should we use when there is a hierarchical classification problem, and why is the proposed metric a more reliable approach than others?

**Limitations:**

The authors state that "One limitation of our general approach is that it may not easily scale to millions of classes, where embeddings can be more suitable." It is not clear why that is the case, why embeddings are more suitable, and what "general approach" really means. Please clarify.

The societal impact is poorly written,  can you clarify what this quote means? "While hierarchical classifiers may mitigate some of the risks of misclassification due to their more logical modes of failure, this may enable more widespread use of automatic classification or lead human users to place excessive trust in the system." It is not clear to me what "logical modes of failure" mean, and I don't see why humans would place excessive trust in the system.

**Strengths And Weaknesses:**


Strengths:

- Hierarchical classification is very interesting to the research community.


Weaknesses:

- There is no theoretical motivation or practical motivation for using "Operating characteristic curve" as a metric for Hierarchical classification, why is it better than the other metrics in terms of reliability, correctness, etc.? For example,  [26] proposed an improved version of the mistake-severity metric that  "could give a wrong impression of improvement while the model might just be making additional mistakes to fool the metric" which is a good motivation for their work. That type of motivation is missing in this work.

- The document is poorly written as it is difficult to find out what the authors actually did. The authors do not clearly present their contributions. I suggest the authors summarize their contributions at the end of the introduction as a paragraph or bullet points.

- The related work is poorly written, there is a large list of existing methods described but the authors do not show how relevant they are to the proposed metric/loss functions. It is important that by the end of related work paragraphs to describe what the paper proposes in relation to existing methods.

- The authors spend so much text explaining previous methods and metrics without explaining why they should be there as they do not help in understanding the proposed loss functions and metrics. A simple citation and few sentence explanations of these methods would be sufficient. This amount of detail just adds confusion about finding out what the authors are actually trying to propose.

- The proposed soft-max-descendant performs poorly in so many of the settings so it is not clear what the motivation is for including it in this work.

---

> ### Author Response · Authors · 2022-08-01
> **Primary response to Reviewer Kenf**
>
> Thanks for your feedback.
> It seems that the negative evaluation stems from the fact that we simply did not state the motivation and contributions clearly enough.
> We believe it is possible to modify the text such that the significance of the work can be recognized.
>
> > "There is no theoretical motivation or practical motivation for using 'Operating characteristic curve' … why is it better than the other metrics?"
>
> For any problem where prediction involves a trade-off, it is vital to evaluate methods at multiple operating points for 2 main reasons:
> 1. Fair comparison.
> If we compare two methods using one operating point each, it might not be an apples-to-apples comparison, and invalid conclusions could be drawn.
> For example, in Figure 1, if we compared *soft-max-margin* and *flat softmax* using majority inference (τ = 0.5, triangle marker), we would conclude that *soft-max-margin* is more specific (x axis) and *flat softmax* is more correct (y axis).
> The curves reveal this to be incorrect: *soft-max-margin* is more correct at the same specificity.
> 2. Application-specific requirements.
> Operating curves enable practitioners to select methods based on individual requirements, such as minimum correctness rate.
>
> This is why ROC and precision-recall curves are universal in binary classification and detection, respectively.
> We will make this more clear in Introduction, paragraph 2.
> Reviewer @CWNt said that "this sounds a must-have for the evaluation".
>
> We emphasize that the operating curves in the paper *all use existing metrics*.
> The key difference is the evaluation protocol, which considers the range of predictions obtained from a threshold-based inference rule, rather than a single prediction from a static inference rule.
>
> > "what is the takeaway? What loss function should we use when there is a hierarchical classification problem?"
>
> From Fig 1, it is trivial to determine that soft-max-margin and DeepRTC-softmax are the most effective methods, as they dominate the others.
> Our OOD experiment suggests DeepRTC-softmax to be more suitable for practical applications due to its robustness to unseen classes.
>
> However, the key takeaway is to always consider the trade-off when evaluating hierarchical classifiers, otherwise the wrong conclusion could be drawn.
>
> > "It's difficult to find out what the authors actually did. … I suggest the authors summarize their contributions at the end of the introduction as a paragraph or bullet points."
>
> We propose to include this list:
>
> * We introduce a novel, efficient technique for evaluating hierarchical classifiers that captures the full trade-off between specificity and correctness using a threshold-based inference rule.
> This enables fairer comparison and better characterization of different methods.
> * We propose two novel loss functions, soft-max-descendant and soft-max-margin, as well as a simple modification of DeepRTC.
> While soft-max-descendant is ineffective, soft-max-margin and (modified) DeepRTC achieve the best results.
> * We conduct an empirical comparison of loss functions and inference rules using the iNat21-mini dataset and its hierarchy of 10,000 species.
> The naive softmax loss outperforms other approaches, with the exception of DeepRTC and our soft-max-margin loss.
> The threshold-based inference rule is shown to be highly effective.
> * The robustness of different methods to unseen leaf-node classes is evaluated, showing that top-down conditional approaches can have an advantage in this setting.
>
> We feel that the contributions are significant and ask that you please re-evaluate the merits of the work.
>
> > Related work
>
> We can remove citations for specialized architectures L64-67, and add a contextualizing sentence each to Hierarchical metrics and Structured prediction.
>
> > "The authors spend so much text explaining previous methods and metrics without explaining why they should be there as they do not help in understanding the proposed loss functions and metrics."
>
> For the paper to be self-contained and reproducible, we simply gave the definition of the inference rules and loss functions being compared in the main experiment, as well as the metrics for which we constructed operating curves.
>
> > Why include soft-max-descendant if it performs poorly?
>
> We retained this negative result to save others the effort, since it seemed like a logical loss to use with this parameterization.
> It can easily be removed or shifted to an appendix if the reviewers agree.
>
> > Embeddings for millions of classes? General approach?
>
> With millions of classes, it's common to query an efficient index of class embeddings rather than exhaustively compare an example to every class with a dense layer (our "general approach").
> This is unclear and we will re-phrase.
>
> > Logical modes of failure? Excessive trust?
>
> We mean: Automation bias is more likely if a system makes less flagrant errors, and hierarchical classifiers make less flagrant errors because they predict superclasses when uncertain.

---

> > ### Comment · Reviewer_Kenf · 2022-08-09
> > **response**
> >
> > Thanks for addressing most of my main concerns, reading the rebuttal and the other reviewers reponses I got a clearer picture of what the contributions are for this work. While the experiments, tradeoffs, and the thought experiments seem interesting,  the results are not that impressive seeing how close (and sometimes worse) than Deep RTC  and Softmax Flat, so there seems to be a big room for improvement. However, I will raise my score to a 6 "Weak Accept", because the work seems interesting enough to be published at a conference like NeurIPS.

---

### Official Review · Reviewer_CWNt · 2022-07-18

**Rating:** 6
**Confidence:** 3
**Soundness:** 3 good
**Presentation:** 4 excellent
**Contribution:** 3 good

**Summary:**

This paper presented a new approach for evaluating hierarchical classifiers for predicting non-leaf nodes at multiple operating points. With the new evaluation metric, the paper shows that the widely used softmax classifier performs reasonably well in many cases, although a structured classifier might have an advantage in classifying unseen classes. The paper also proposes two new loss functions. One of them outperforms the softmax classifier in all examinations. The experiments are on ImangeNet and iNat classification tasks.

**Questions:**

Does Eq-17 have any limitations? For example, how was its accuracy in the most popular classification setting that only predicts leaf classes?

**Limitations:**

Lines 340-344 have the discussion and make sense to me.

**Strengths And Weaknesses:**

The paper is well written and easy to follow. A large amount of literature is discussed and adequately summarized. The comparison between the paper's and previous methods is succinct but informative.

Regarding significance, comparing hierarchical classifiers at multiple operating points sounds a must-have for the evaluation. Therefore I am slightly surprised that the paper mentioned that "almost no prior work has compared methods at multiple operating points" (in lines 5 and 6). If the above claim is valid, I believe this work provides a valuable evaluation of the problem and is a significant contribution. I like to admit that I am not an expert on this topic and will cross-check with other reviewers for the claim. To strengthen the significance, the author may explain why evaluation at multiple operating points is not popular or feasible without the method in section 5.

Technically, I do not find obvious mistakes, but the clarity of the key methods may have room to improve. For example, (1) elaborate more on why Eq-17 gives a result better than flat in Figure 1; (2) use an algorithmic block to summarize the process of section 5.

Regarding the experiments, the paper uses large-scale datasets, has proper baselines and prior methods, and examines interesting aspects such as flat classifiers at different levels and unseen classes. The results provide conclusive insights and are easy to interpret.

---

> ### Author Response · Authors · 2022-08-01
> **Primary response to Reviewer CWNt**
>
> Thanks for your review.
> We're glad that you appreciate the importance of evaluation using multiple operating points.
>
> > I am slightly surprised that … "almost no prior work has compared methods at multiple operating points" … If the above claim is valid, I believe this work provides a valuable evaluation of the problem and is a significant contribution. … I am not an expert on this topic and will cross-check with other reviewers for the claim
>
> We were similarly surprised.
> It seems that none of the reviewers have contested this claim.
>
> > To strengthen the significance … explain why evaluation at multiple operating points is not popular or feasible without the method in section 5
>
> Without the method in Section 5, one must repeat the inference procedure (eq. 4) and evaluate the metrics for every threshold.
> If we let Y denote the number of nodes in the hierarchy, then this takes at least O(Y) time.
> Doing this for a resolution of T different thresholds takes O(T Y) time.
> However, the Pareto set, which gives the predictions for *all* thresholds, can be identified in O(Y log Y) time.
> Our algorithm generates the curves with perfect (continuous) resolution and is faster by a factor of T / log Y.
> The naive algorithm takes significantly longer to generate even moderate-resolution plots (e.g. 20 thresholds).
>
> > Does Eq-17 have any limitations? For example, how was its accuracy in the most popular classification setting that only predicts leaf classes?
>
> For iNat21-mini, soft-max-margin (eq. 17) is slightly better than flat softmax classification for the classical task.
> This can be read from Figure 1 (right): the operating curves include leaf inference as a special case (τ = 0, circle marker), and the "Correct" and "Exact" metrics reduce to classical accuracy when the predicted and ground-truth classes are leaf nodes.
>
> Eq. 17 does have the limitation that it is necessary to choose the desired margin, and a poor choice may lead to worse results.
> For example, in Figure 4 (Appendix A), it can be seen that setting the scale too large (α = 20) results in diminished accuracy.
>
> > elaborate more on why Eq-17 gives a result better than flat in Figure 1
>
> Eq. 17 is more effective because, unlike the flat softmax loss, it demands greater separation (in terms of logits) of the true label from the incorrect classes than from the other correct classes.
> This will be added to the text.
>
> > use an algorithmic block to summarize the process of section 5
>
> We agree that this would improve the clarity of Section 5.
> We have written algorithms for two subroutines, `OrderedParetoSet` and `ConstructDatasetCurve`, which will be added to the supplementary material.
> These can be provided during the discussion if desired.
> See also the response to reviewer @53HE concerning this section.

---

> > ### Comment · Reviewer_CWNt · 2022-08-04
> > **Re: response**
> >
> > Thanks for the responses. Yes, please provide the two algorithmic routines (OrderedParetoSet and ConstructDatasetCurve). I have very low confidence about whether my imagination of these routines is the same as your actual process. If possible, please also provide a toy example (ex: a task with only 3 leaf classes and 10 or fewer data points) to demonstrate the outputs of key steps in the routine. A concrete example will significantly reduce our effort in the evaluation.

---

> > > ### Author Response · Authors · 2022-08-06
> > > **Algorithmic blocks and toy example**
> > >
> > > ### Algorithmic blocks
> > >
> > > The first algorithm, `OrderedParetoSet`, is not claimed as a novel contribution; it was introduced in Algorithm 3.1 of [Kung et al. (1975)](http://www.eecs.harvard.edu/~htk/publication/1975-jacm-kung-luccio-preparata.pdf).
> > > However, this ordered set forms the basis of our own algorithm, and we therefore include it for understanding.
> > >
> > > Whereas the original algorithm made the simplifying assumption that no two vectors have the same x or y value, we present a variant that supports duplicate values in either co-ordinate.
> > > This is important in our case, since it is common for many nodes in the hierarchy to have the same information (e.g. all leaf nodes).
> > >
> > > Type annotations: We use `[float]` to denote a list of floats (and thus `[[float]]` to denote a list of lists of floats).
> > >
> > > ```
> > > OrderedParetoSet(x: [float], y: [float])
> > >   // Returns list of indices defining the Pareto set,
> > >   //   ordered such that x is decreasing and y is increasing.
> > >   // Require: len(x) = len(y) = number of points n
> > >   n <- len(x)
> > >   pi <- argsort(y, 'desc')
> > >   x, y <- x[pi], y[pi]
> > >   rho <- argsort(x, 'desc')  // Must be stable sort
> > >   x, y, pi <- x[rho], y[rho], pi[rho]
> > >   // Have lexicographic order:
> > >   // if i < j, then x[i] ≥ x[j] (and if x[i] = x[j], then y[i] ≥ y[j]).
> > >   // Therefore element j is dominated iff ∃ i < j with y[i] > y[j]
> > >   y_max <- cummax(y)  // Like cumsum
> > >   subset <- [j in 0..n-1 where j = 0 or y[j] > y_max[j-1]]
> > >   pi <- pi[subset]
> > >   return pi
> > > ```
> > >
> > > The second algorithm, `ConstructDatasetCurve`, takes a dataset of examples with groundtruth label `y[i]` and predicted distribution `p[i]` (a vector of probabilities) and returns the piecewise-constant function that results as the threshold is decreased from 1 to `tau_min`.
> > > ```
> > > ConstructDatasetCurve(y: [int], p: [[float]], tau_min: float)
> > >   // Returns thresholds C and metric values Z defining piecewise-constant curve.
> > >   // C is monotonically decreasing with C[0] = 1, C[-1] = tau_min.
> > >   // Metric value is Z[j] for C[j] ≥ tau > C[j+1].
> > >   // Inputs: For example i, groundtruth label y[i] and probability vector p[i],
> > >   //   minimum threshold min_tau.
> > >   // Require: len(y) = len(p) = number of examples N
> > >   // Require: len(p[i]) = number of nodes Y
> > >   // Require: y[i] in [0, Y), p[i][y] in [0, 1], tau_min in [0, 1)
> > >   // Require: p[i][0] = 1 (root node)
> > >   // External: Constant vector I[y] with len(I) = Y
> > >   // External: Function M(y, y_hat) that evaluates metric
> > >   N <- len(y)
> > >   Z_0 <- 0
> > >   C <- []
> > >   Delta <- []
> > >   for i in 0..N-1  // For each example
> > >     subset <- [y in 0..Y-1 where p[i][y] > tau_min]  // Nodes that satisfy min threshold
> > >     pi <- OrderedParetoSet(p[i][subset], I[subset])
> > >     y_hat <- subset[pi]  // Possible predictions for any tau ≥ tau_min
> > >     K <- len(y_hat)
> > >     z <- [M(y[i], y_hat[k]) for k in 0..K-1]  // Evaluate metric for each prediction
> > >     delta <- diff(z)
> > >     c <- p[i][y_hat[1:]]
> > >     Z_0 <- Z_0 + z[0]
> > >     C <- concat(C, c)
> > >     Delta <- concat(Delta, delta)
> > >   end for
> > >   order <- argsort(C, 'desc')  // Note: Can use priority queue for slightly faster merge
> > >   C, Delta <- C[order], Delta[order]
> > >   C <- concat([1], C, [min_tau])
> > >   Delta <- concat([0], Delta)
> > >   Z <- (1/N) * (Z_0 + cumsum(Delta))
> > >   return C, Z
> > > ```
> > > Above, we consider a single, scalar metric `M(y, y_hat)`.
> > > In practice, we simultaneously consider at least 2 metrics (e.g. recall, precision) by having `M(y, y_hat)` return a vector of metrics.
> > > This makes the "delta" and "z" variables be vector sequences instead of scalar sequences.
> > >
> > > ### Toy example
> > >
> > > As an example, let's consider a basic hierarchy with 5 nodes:
> > > ```
> > >   0
> > >  / \
> > > 1   2
> > >    / \
> > >   3   4
> > > ```
> > > with a uniform prior on the leaves: `I(1) = I(3) = I(4) = -ln(1/3) ≈ 1.10`, `I(2) = -ln(2/3) ≈ 0.41`, `I(0) = -ln(1) = 0`.
> > >
> > > Now consider 2 examples (A and B), both with groundtruth `y = 3`, and predictions
> > > ```
> > > A: p(0) = 1, p(1) = 0.1, p(2) = 0.8, p(3) = 0.6, p(4) = 0.1
> > > B: p(0) = 1, p(1) = 0.2, p(2) = 0.7, p(3) = 0.2, p(4) = 0.3
> > > ```
> > > Example A has the correct leaf prediction, example B does not.
> > > The Pareto sets for these examples (from `OrderedParetoSet`) are:
> > > ```
> > > A: y_hat = [0, 2, 3], p(y_hat) = [1, 0.8, 0.6], I(y_hat) = [0, 0.41, 1.10]
> > > B: y_hat = [0, 2, 4], p(y_hat) = [1, 0.7, 0.3], I(y_hat) = [0, 0.41, 1.10]
> > > ```
> > > The `ConstructDatasetCurve` function evaluates the metric for all predictions, computes the metric deltas, merges the lists and takes the partial sum.
> > > Using the "Recall" metric and `min_tau = 0`, we have:
> > > ```
> > > A: z = [0, 0.41, 1.10]/1.10 = [0, 0.37, 1], delta = [0.37, 0.63], c = [0.8, 0.6]
> > > B: z = [0, 0.41, 0.41]/1.10 = [0, 0.37, 0.37], delta = [0.37, 0], c = [0.7, 0.3]
> > > Z_0 = 0
> > > // merge c's and deltas; sort descending by c
> > > C = [0.8, 0.7, 0.6, 0.3]
> > > Delta = [0.37, 0.37, 0.63, 0]
> > > // prepend/append initial/final value
> > > C = [1, 0.8, 0.7, 0.6, 0.3, 0]
> > > Delta = [0, 0.37, 0.37, 0.63, 0]
> > > Z = (1/2) * (Z_0 + cumsum(Delta)) = [0, 0.18, 0.37, 0.68, 0.68]
> > > ```
> > > This gives piecewise-constant recall: it increases from 0 (for 1 > tau ≥ 0.8) to 0.68 (for 0.6 > tau ≥ 0).

---

### Author Response · Authors · 2022-08-09
**Updated PDFs**

We have uploaded a revised version of the paper to address the main requests. Here is a summary of the changes.

* Change paragraph 2 of Introduction to make motivation of curves more explicit
* Add bullet point contributions at end of introduction
* Related work: Remove citations that are not directly related; make connection to this paper more explicit
* Revise Section 5 for clarity; use piecewise-continuous functions and add equation showing summations
* Explain meaning of particular C(y, y') for Conditional Risk Minimization (Inference functions) and for soft-max-margin (Loss functions)
* Fix mistakes in definition of C(y, y')
* Re-word "general approach" in context of scaling to many classes (Conclusion)
* Re-word "logical modes of failure" and "excessive trust" (Social impact)

And in the supplementary material there are two new appendix sections:

* Table giving all parameterizations and their different properties
* Two algorithmic blocks for Section 5

---

### Meta-Review · Area_Chair_6gMV · 2022-08-25

**Recommendation:** Accept
**Confidence:** Certain

**Metareview:**

The submission benchmarks several hierarchical classification techniques showing that flat softmax on the leaf nodes dominates most methods.  This is quite a negative result for previous works, indicating that efforts on hierarchical classification losses are often not leading to better results.  The authors introduce a loss function that does appear to result in  performance beating the flat classification baseline.  The writing style flows, there is a reasonably good review of the literature, and results appear to give some insights into methods for performing and evaluating hierarchical classification methods, which are complementary to the existing literature on a problem that has been studied in various forms for decades.  The reviewers were unanimous in their opinion that the submission is just over the threshold for acceptance at NeurIPS after the rebuttal process.

**Award:**

No

---

### Decision · Program_Chairs · 2022-09-14

Accept